TECHNICAL RELEASE

# TSTA: thread and SIMD-based trapezoidal pairwise/multiple sequence-alignment method

Peiyu Zong[1,2,†], Wenpeng Deng[1,2,†], Jian Liu[2] and Jue Ruan[2,*]

1 Hubei Key Laboratory of Agricultural Bioinformatics, College of Informatics, Huazhong Agricultural University, Wuhan, 430070, China

2 Shenzhen Branch, Guangdong Laboratory of Lingnan Modern Agriculture, Genome Analysis Laboratory of the Ministry of Agriculture and Rural Affairs, Agricultural Genomics Institute at Shenzhen, Chinese Academy of Agricultural Sciences, No 7, Pengfei Road, Dapeng District, Shenzhen, 518120, Guangdong, China

## ABSTRACT

The rapid advancements in sequencing length necessitate the adoption of increasingly efficient sequence alignment algorithms. The Needleman–Wunsch method introduces the foundational dynamic-programming matrix calculation for global alignment, which evaluates the overall alignment of sequences. However, this method is known to be highly time-consuming. The proposed TSTA algorithm leverages both vector-level and thread-level parallelism to accelerate pairwise and multiple sequence alignments.

**Availability and implementation:** Source codes are available at https://github.com/bxskdh/TSTA.

**Subjects** Software and Workflows, Bioinformatics, Genetics and Genomics

**Submitted:** 06 June 2024

* Corresponding author. E-mail: ruanjue@caas.cn

† Contributed equally.

Preprint submitted at https://doi.org/10.1101/2024.09.18.613655

## INTRODUCTION

Sequence alignment is a crucial foundation of bioinformatics, playing a vital role in the analysis of biological sequence data. Currently, the Smith–Waterman algorithm [1] and the Needleman–Wunsch algorithm [2] are the most commonly used for sequence comparison. These algorithms calculate the maximum score for each cell in a dynamic programming (DP) matrix and track the optimal source path. Gotoh [3] later improved these algorithms by introducing an affine gap penalty, which extends the linear gap penalty used in the original algorithms and allows for a more focused gap representation. Although all three algorithms succeed in achieving optimal alignment results, they require significant computational time when applied to lengthy sequences. Various optimization techniques have been developed to enhance the efficiency of sequence alignment.

### Single instruction multiple data (SIMD)

A highly effective approach to accelerating the computation of scoring matrices is to employ vectorized computing for independent cells. This method has become more practical with the advent of SIMD computing, which supports spatial parallelism by executing multiple data operations simultaneously using a single instruction. Prior to this advancement, Alpern and Bowen [4] recommended using microparallelism for protein matching. This approach is akin to SIMD technology in that it allows for the simultaneous alignment of a query protein sequence with multiple reference protein sequences, utilizing the same

control structure at the instruction level and enabling parallel sequence-to-bank alignment. In a different approach, Wozniak [5] rearranged each row of the initial scoring matrix uniquely, aligning diagonal cells into columns, thereby facilitating the parallel acceleration of sequence-to-sequence operations through a SIMD-like Visual Instruction Set technology. Conversely, Rognes *et al.* [6] transformed the SIMD method from a diagonal to a parallel configuration relative to the query sequence. Their algorithm leverages MultiMedia eXtensions with a 64-bit single instruction and Streaming SIMD Extensions (SSE) with a 128-bit instruction. Subsequent algorithms have integrated advanced vector extensions (AVX) techniques, such as libssa [7], and novel 512-bit SIMD instruction sets (AVX-512), exemplified by XSW [8].

## Difference recurrence relation

In the SIMD technique, registers have a fixed length and can hold multiple types of data. By utilizing smaller data types, it becomes possible to perform computations on a larger number of values simultaneously. Initially, Myers [9] employed a relocatable DP matrix that includes both horizontal and vertical deltas in the Fast Bit-Vector Algorithm for Approximate String Matching. This algorithm is used for approximate string matching. Shortly thereafter, BitPAl [10] utilized a similar technique for sequence alignment, integrating a more comprehensive scoring methodology that relies on three distinct values: match, mismatch, and gap. BitPAl demonstrates the potential for further optimization using the SIMD approach. Suzuki and Kasahara [11] integrated and optimized incremental relations and SIMD techniques in dynamic banded DP (adaptive banded DP), referring to the former as difference. This method employs the SIMD technique to calculate difference recurrence relations (DRR) in both horizontal and vertical directions for specific cells on the antidiagonal. By reducing the stored values in the standard DP matrix from 32-bit to 8-bit, the number of cells computed simultaneously increases by a factor of 4. Minimap2 (RRID:SCR_018550) [12] extends the DRR to a two-piece affine gap cost-model and, to implement an SSE technology more efficiently on this basis, it transforms row-column coordinates into diagonal-antidiagonal coordinates.

## Striped SIMD

Wozniak's method [5] demonstrates that loading values on the reverse diagonal of the DP matrix requires an additional complex process due to the discontinuous nature of these values during storage. Rognes *et al.* [6] found that vectorizing cells along the query sequence results in faster processing. However, it still necessitates non-vectorized computations for resolving dependencies. Subsequently, Farrar [13] proposed a technique called "striping", wherein cells are reorganized along the query sequence into multiple strips. This method resolves dependency relationships by incorporating several correction cycles in the direction of the dependency. The number of correction loops is determined by the length of the gap. This approach has been employed in numerous subsequent algorithms, such as SWPS3 [14], CUDASW++2.0 (RRID:SCR_008862) [15], and others. Some mappers also employ striped SIMD techniques, such as BWA-MEM (RRID:SCR_022192) [16], which uses this technique to realign paired-end reads that may have been unmapped for various reasons. Recently, bsalign [17] optimized the correction method in the F direction through an active and proactive "breakdown" approach.

## Multithreading

An alternative method to accelerate the computation process is to employ multiple threads for parallel calculation of the non-dependent sections of the sequence alignment. This approach is efficient when the benefit of speeding up these threads outweighs the cost of additional overhead. The multithreaded technique can significantly enhance inter-sequence alignments, particularly when numerous alignments are required. Rognes' SWIPE software (RRID:SCR_012771) [18] utilizes this inter-sequence parallel approach for the Smith–Waterman algorithm, achieving outstanding results for local alignments. The Parasail library (RRID:SCR_021805) [19] extends the range of alignment by developing an independent C library that includes global, semi-global, and local alignment. The BGSA [20] algorithm employs data-level parallelism using multiple processing units, such as CPUs and Xeon Phis, and applies them in bit-parallel global sequence-alignments. Additionally, Jacek Blazewicz *et al.* [21] enhanced the progressive multiple sequence alignment method using the inter-sequence parallel approach. This involves dividing the task matrix into smaller windows, each processed on a single GPU, where individual threads align pairs of sequences within each window. In contrast, the intra-sequence thread-parallelism technique was initially utilized by Edmiston *et al.* [22] and Liu *et al.* [23], who applied multithreading to non-dependent cells located on the anti-diagonal in the scoring matrix. Rahn *et al.* [24] introduced the Generalised Wavefront Model to group certain independent blocks in the score matrix, such as blocks near the anti-diagonal, based on the instruction set type. Subsequently, the blocks within each group were computed vertically using the SIMD instruction set.

## POA

In 2002, the field of multiple sequence alignment (MSA) research saw the introduction of a novel approach called partial order alignment (POA) [25], which employed graph-based techniques. POA uses directed acyclic graphs to establish connections between sequences. Unlike in pairwise sequence alignment (PSA), the source of a node in POA is not restricted to a single node during alignment. Nodes are merged based on their alignment scores, and the final alignment result is derived from the fused graph. Similarly, the SIMD approach can be integrated into the POA algorithm. Vaser *et al.* developed the SPOA [26] algorithm, which uses SIMD vectors to parallelize computations along the query sequence. It performs parallel computations that are both perpendicular to the query sequence and along the diagonal line. However, due to data dependency, horizontal computations are performed linearly rather than using SIMD operations for efficiency. Additionally, the abPOA method developed by Yan Gao *et al.* [27] integrates the adaptive banded DP approach into the SPOA algorithm. This incorporation effectively reduces the computational memory and time required for computation.

This paper presents an accelerated algorithm for the global alignment of pairwise and multiple sequences. The algorithm integrates four methods: the difference method, the stripe method, the SIMD instruction set, and multithreading. These integrations result in significant improvements in the speed of both pairwise and multiple sequence alignment, particularly for long sequences.



## MATERIALS AND METHODS

### Global alignment

In contrast to local alignment, which seeks to identify local overlaps, global alignment evaluates entire sequences to determine the optimal alignment. The Needleman–Wunsch–Gotoh algorithm is employed for global alignment scoring:

$$M_{(0,0)} = 0, \quad E_{(0,i)} = \text{INT\_MIN}, \quad F_{(j,0)} = \text{INT\_MIN}$$

Initial:

$$M_{(i,j)} = \max \begin{cases} E_{(i,j)} \\ F_{(i,j)} \\ M_{(i-1,j-1)} + D \end{cases} \tag{1}$$

Extension:

$$E_{(i,j)} = \max \begin{cases} E_{(i-1,j)} + e \\ M_{(i-1,j)} + e + o \end{cases}$$
$$F_{(i,j)} = \max \begin{cases} F_{(i,j-1)} + e \\ M_{(i,j-1)} + e + o \end{cases} \tag{2}$$

In this context, $M_{(i,j)}$ represents the optimal score for the coordinates $(i, j)$ in the scoring matrix, whereas $E_{(i,j)}$ and $F_{(i,j)}$ represent the scores in the vertical and horizontal directions, respectively. The variable $e$ denotes the penalty score for extending a gap, while the variable $o$ represents the penalty score for opening a gap. The variable $D$ stands for the match/mismatch score. This algorithm utilizes an affine penalty model.

### A brief overview of the POA

The POA method converts the sequence into a DAG, where each base in the sequence is represented as a node in the graph. Directed edges, indicating dependencies between nodes, represent their connections (Figure 1A). Subsequently, the nodes are arranged in a topological order, resulting in a linear-like structure. Following this arrangement, a computation resembling the pairwise sequence alignment model occurs, albeit with the distinction that the score of each cell is not exclusively derived from three sources (Figure 1B). Upon completion of the score matrix calculation, nodes with matching sequences are merged to form a new graph (Figure 1C), which then undergoes iterative calculations for subsequent sequences.

### Difference method

The difference method is implemented as an alternative to the classic Needleman–Wunsch–Gotoh alignment algorithm, facilitating parallel computation of substantial data volumes through the utilization of registers in the SIMD instruction set, capable of accommodating varying bit sizes. This approach employs $\Delta H$ and $\Delta V$ (Figure 2) to denote the difference value (*D*-value) between adjacent cells in horizontal and vertical orientations. Typically, substantial enhancements in parallel efficiency can be achieved by leveraging the minimal specification of 8 bits in the registers.



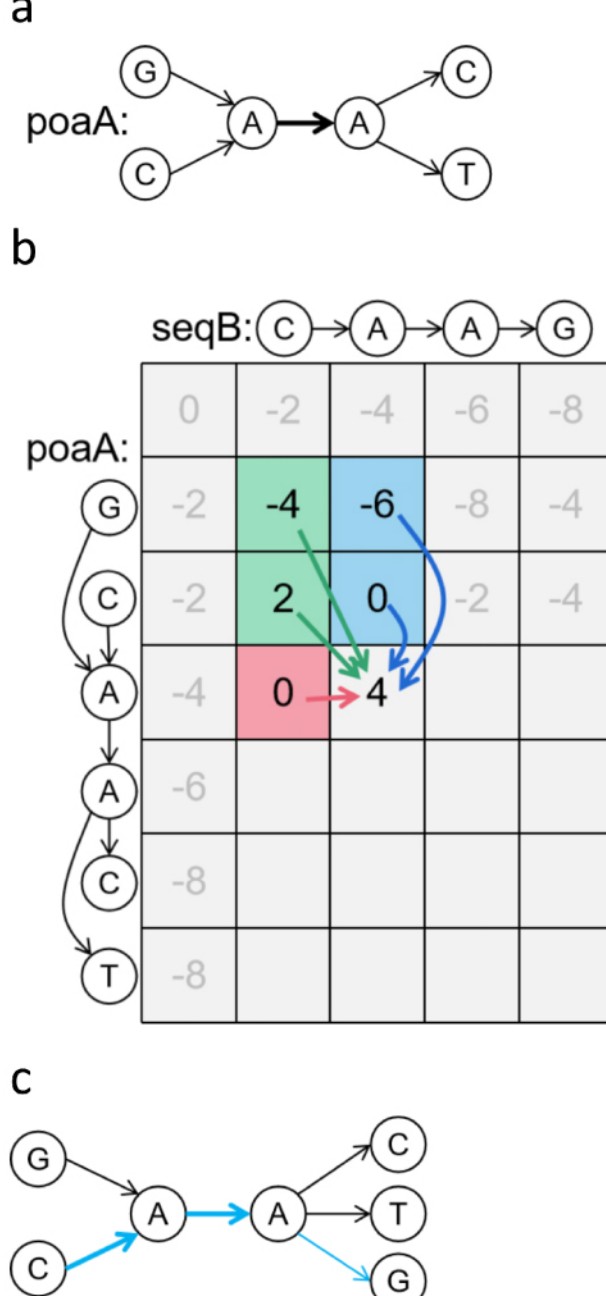

**Figure 1.** (A) POA graph and sequences of the alignment. The bold path indicates that multiple sequences have passed through. (B) The scoring matrix. The POA graph is topologically sorted into a linear-like structure and aligned with seqB. The scores of cells (A, A) are derived from the maximum values in five directions. (C) The new POA graph with the seqB sequence is in blue.

PSA initially calculates the maximum $D$-value S between the current cell and the top left cell. It next calculates the horizontal $D$-value $\Delta H$ and vertical $D$-value $\Delta V$ of the current

**a**

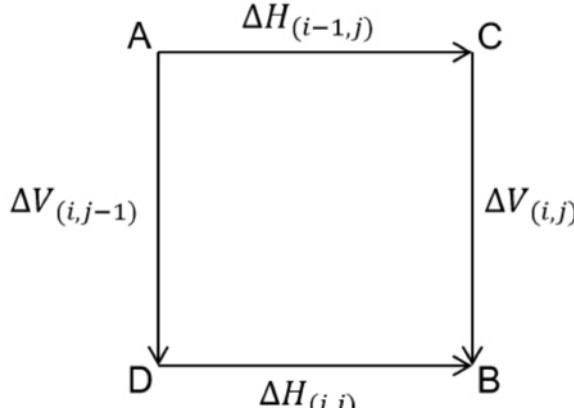

**b**

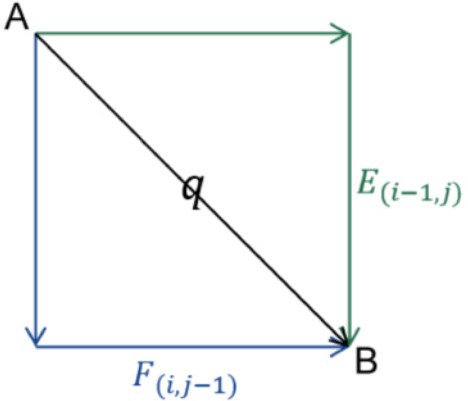

**Figure 2.** The difference method of pairwise sequence alignment. $\Delta H$, $\Delta V$ respectively represent the difference between adjacent cells in the horizontal and vertical directions. $E$ is the difference between the score of B obtained in the vertical direction and the score of A, and $F$ is the difference between the score of B obtained in the horizontal direction and the score of A. The variable $q$ is the matching/mismatch score.

cell individually:

$$S_{(i,j)} \;=\; \max \begin{cases} E_{(i-1,j)} \\ q \\ F_{(i,j-1)} \end{cases} \tag{3}$$

$$E_{(i,j)} \;=\; \max \begin{cases} E_{(i-1,j)} + e \\ S_{(i-1,j)} + o + e \end{cases} - \Delta V_{(i,j-1)} \tag{4}$$

$$F_{(i,j)} \;=\; \max \begin{cases} F_{(i,j-1)} + e \\ S_{(i,j-1)} + o + e \end{cases} - \Delta H_{(i-1,j)} \tag{5}$$

$$\Delta V_{(i,j)} \;=\; S_{(i,j)} - \Delta H_{(i-1,j)} \tag{6}$$

$$\Delta H_{(i,j)} \;=\; S_{(i,j)} - \Delta V_{(i,j-1)}. \tag{7}$$

MSA directly calculates the maximum horizontal *D*-value $\Delta H$ of the current cell, and then calculates the $\Delta V_x$ between the current cell and each previous node:

$$\Delta H_{(i,j)} \quad = \quad \max \begin{cases} \max(E_{1(i-1,j)}, F_{1(i,j-1)}, q) - \Delta V_{1(i,j-1)} \\ \max(E_{2(i-1,j)}, F_{2(i,j-1)}, q) - \Delta V_{2(i,j-1)} \\ \qquad\qquad \dots \end{cases} \tag{8}$$

$$E_{(i,j)} \quad = \quad \max \begin{cases} E_{1(i-1,j)} + e - \Delta V_{1(i,j-1)} \\ E_{2(i-1,j)} + e - \Delta V_{2(i,j-1)} \\ \qquad\quad \dots \\ \\ \Delta H_{(i,j)} + o + e \end{cases} \tag{9}$$

$$F_{x(i,j)} \quad = \quad \max \begin{cases} F_{x(i,j-1)} + e \\ \Delta H_{(i,j)} + o + e + \Delta V_{x(i,j-1)} \end{cases} - \Delta H_{x(i-1,j)} \tag{10}$$

$$\Delta V_{x(i,j)} \quad = \quad \Delta H_{(i,j)} + \Delta V_{x(i,j-1)} - \Delta H_{x(i-1,j)}. \tag{11}$$

The computational approaches for PSA and MSA exhibit slight variations. The PSA method computes the maximum *D*-value between the upper left cell and the current cell, and subsequently subtracts $\Delta V_{(i,j-1)}$ to obtain $\Delta H_{(i,j)}$. Conversely, the MSA method capitalizes on the uniqueness of $\Delta H_{(i,j)}$ to determine its maximum value (Figure 3).

## Stripe method

Farrar [9] recommends employing a stripe method to enhance the implementation of in-sequence parallelism, resulting in significant improvements in parallel performance through the utilization of the SIMD instruction set. This method involves reorganizing the originally sequential computation. The length of each stripe corresponds to the processing capacity of the SIMD register, with any remaining space in the register filled with dummy symbols having no impact on the outcome. Farrar's approach utilizes the Lazy-F method in parallel computing to eliminate data interdependencies among F vectors, whereas the TSTA algorithm employs the Active-F method. The former necessitates multiple updates to the value of each register after the initial loop until the ultimate value is achieved, while the latter updates the value of the last register obtained after the initial loop and subsequently executes another loop to attain the ultimate value (Figure 4).

Through the integration of the difference approach with the stripe method, it becomes feasible to extend the lengths of individual stripes, such as SSE (16), AVX (32), and AVX512 (64). This strategy can be employed in both PSA and MSA.

## Thread-level ladder parallelism

Among the in-sequence parallelism methods, calculations utilizing an anti-diagonal layout were among the earliest to be conceived and are frequently utilized. While this method is not compatible with the SIMD instruction set parallelism, employing threads to compute independent units on the anti-diagonal remains a viable option. However, allocating computational resources to invoke threads constitutes a significant portion of the total runtime. Therefore, employing threads solely to compute each cell on the anti-diagonal does not yield favorable results. Consequently, the TSTA algorithm divides the entire scoring matrix into multiple blocks of equal length and width along the anti-diagonal. These blocks are then computed in parallel threads to substantially minimize the overhead

a

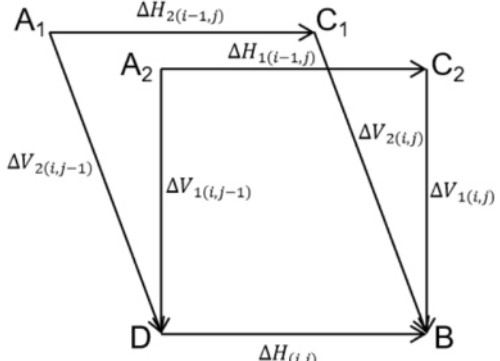

b

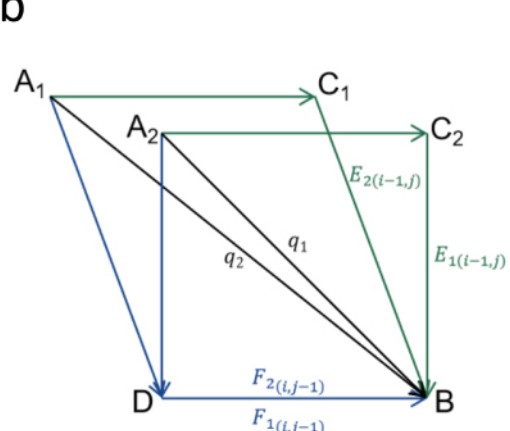

**Figure 3.** The difference method of POA multiple sequence alignment. The variables $\Delta H_1$ and $\Delta H_2$ represent the differences between neighboring cells in the horizontal direction of the two front nodes in the POA graph. Similarly, the variables $\Delta V_1$ and $\Delta V_2$ represent the differences between adjacent cells in the vertical direction of the two front nodes in the POA graph. Due to the existence of two preceding nodes, there will be a total of two instances of $E$ and $F$. Because $q$ is only related to whether the base matches, the values of $q_1$ and $q_2$ are equal.

associated with invoking threads. Within each block, a parallel technique is employed, combining the difference method and the stripe method to enhance parallel efficiency.

The formula utilized in this method is $L = S \times D$, where $L$ represents the width of the block in bits, $S$ represents the size of the register (16/32/64 bits), and $D$ is the number of stripes. Both the values of $S$ and $D$ influence the speed of the calculation, though the value of $S$ is contingent upon the machine's level of support.

This research entails the incorporation of all blocks along each anti-diagonal into the thread pool as tasks. Subsequently, the calling threads within the thread pool engage in competition for the independent completion of these tasks individually (Figure 5). It is ensured that the block tasks on the subsequent anti-diagonal are not incorporated into the thread pool until all blocks on the current anti-diagonal have been computed, accounting for their dependencies.

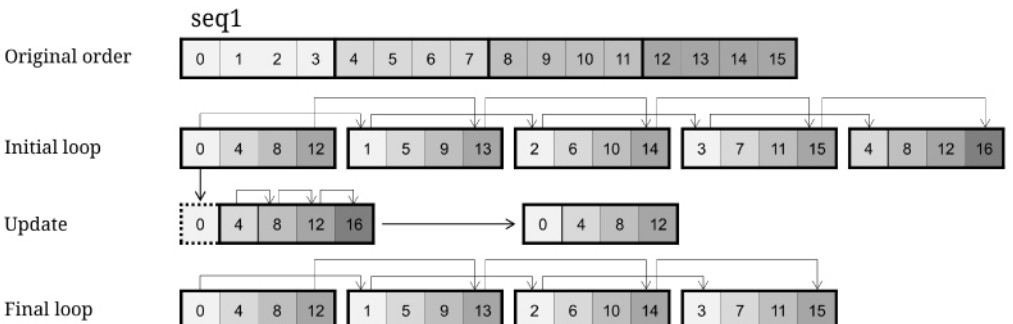

**Figure 4.** Stripe method. **Original order**: The original sequence, length = register size × number of stripes. **Initial loop**: Rearranges the original sequence by grouping every four cells in a striped pattern, resulting in a non-sequential arrangement. Then, it proceeds with the initial loop computation. Due to the correctness of the first value in the first register and the consecutive nature of the calculations, the first value in each register is correct. **Update**: Updates the value in the last register (excluding the first value) acquired from the initial loop. Each score is derived from three sources (multiple sequence alignment can also be generally categorized into three types of sources). The vertical and match/mismatch scores are already known, and the first value of the register is confirmed to be precise. Thus, in this stage, it solely assesses if the horizontal score has been breached. If it has been breached, it will be updated; otherwise, it will remain unaltered. **Final loop**: Performs a second loop based on the updated values to obtain the final correct value.

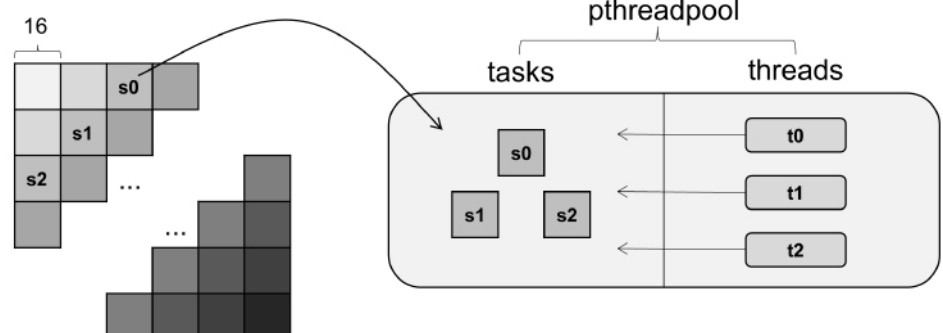

**Figure 5.** Thread parallel method. First, it partitions the entire matrix into multiple anti-diagonal lines, with each line consisting of independent blocks of equal size. Subsequently, it allocates all blocks situated on each diagonal to the task pool of the thread pool, where the thread concurrently computes the score of these blocks. The block in the picture has a width of 16 bases.

To minimize memory usage, each thread in PSA only requests a memory block of char type that is two units long (16 × 2, as illustrated in Figure 4) to store the $\Delta H$ values. This memory block is released once the block has been processed. The algorithm utilizes a single memory to store and update the true value of the vertical total length of the scoring matrix, which is also employed to determine the maximum value of the matrix and the necessary $\Delta V$ value for each thread calculation. Additionally, a dedicated memory is utilized to store and update the $\Delta H$ value of the horizontal total length of the scoring matrix, with each thread accessing and modifying this value from memory. These two memories alternate as dependency rows and computation rows. Moreover, the last true value of each row and the $\Delta H$ value of the last row are recorded to calculate the related blocks of the next anti-diagonal line. In the context of MSA, nodes that do not span block boundaries necessitate a memory allocation equivalent to the block length to store the $\Delta H$ value.

Conversely, nodes that cross block boundaries require a memory allocation equal to the length of the sequence to store $\Delta H$ values.

## RESULT

The test was divided into two groups based on the presence of backtracking regarding PSA. Within the context of backtracking, TSTA was evaluated alongside bsalign [13] and parasail. Conversely, the group without backtracking compared TSTA with align_bench_wave [22] and parasail. In addition, TSTA was compared to SPOA, abPOA, and bsalign for MSA.

Across all evaluations, we employed the affine gap model, featuring a match score of 2, a mismatch score of −3, and gap extension and gap open penalties of −2 and −4, respectively. The align_bench_wave algorithm utilized its inherent scoring system. All experiments, except for abPOA, were conducted with the band disabled and set to global alignment.

The experiments were conducted on a computer equipped with 2.10 GHz Intel Xeon Gold 6230R CPUs, featuring 52 cores and 376 GB of RAM. We evaluated the performance of our algorithm, SPOA, and bsalign using the SSE4.2 instruction set. Additionally, we assessed the performance of abPOA and parasail using the AVX2 instruction set and align_bench_wave using the AVX-512 instruction set. Notably, the SPOA software is installed using Conda (RRID:SCR_018317), exhibiting a faster alignment speed than the AVX-optimized version included in the git repository. Furthermore, the parasail library employs a 32-bit integer data type to mitigate the risk of computational overflow. We utilized 30 threads for align_bench_wave, parasail, and our algorithm.

In the aforementioned algorithms, we assessed the time expended on both real and simulated datasets. The real datasets comprised three sets of sequences with average lengths of 1k bps (accession number ERR12439694), 10k bps (accession numbers ERR11890836 and SRR10673221), and 50k bps (accession numbers SRR27371247 and SRR27371637), respectively. The datasets in PSA were categorized into 637, 8,600, and 17 alignments. Meanwhile, the authentic data in MSA were categorized into groups of five sequences (5×) and ten sequences (10×). Within the 5× groups, there were 254, 3,399, and 7 group alignments, whereas in the 10× groups, there were 127, 1,699, and 3 group alignments. The data source can be viewed in the DATA AVAILABILITY section [28].

For the simulated dataset, we randomly selected 100 sequences of 200k bases' regions from GRCh38. Subsequently, we employed PbSim2 (RRID:SCR_002512) [29] to simulate query sequences for each reference region. One query sequence was chosen from each set to serve as input sequence along with its respective reference sequence for PSA. Furthermore, subsets of five and ten sequences were selected from the query sequences to form two sets of input sequence groups for MSA. Due to the decreased efficiency of MSA, particularly with larger sequences (>100k bases), varied sequence lengths were utilized in testing. Additionally, the maximum memory limit of the computer restricts the sequence length that can be processed, whereas PSA is more flexible in this regard. In the context of PSA, sequence lengths were set at 10k, 20k, 50k, 100k, and 200k. Moreover, in the context of MSA, the number of sequences was set at 5 and 10, with lengths of 5K, 10K, 20K, and 50K.

## Evaluating time with different numbers of threads and stripes

The number of threads is denoted by $T$, while the number of stripes in the striping method is represented by $B$. We assessed the optimal outcomes for various combinations of $T$ and $B$ values. Figure 6 illustrates the speed for several combinations at a rate of pairwise

a

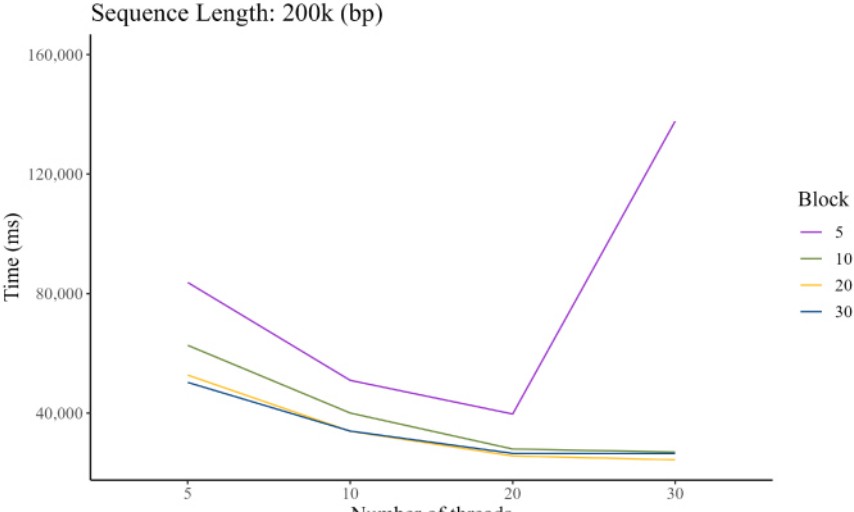

Sequence Length: 200k (bp)

b

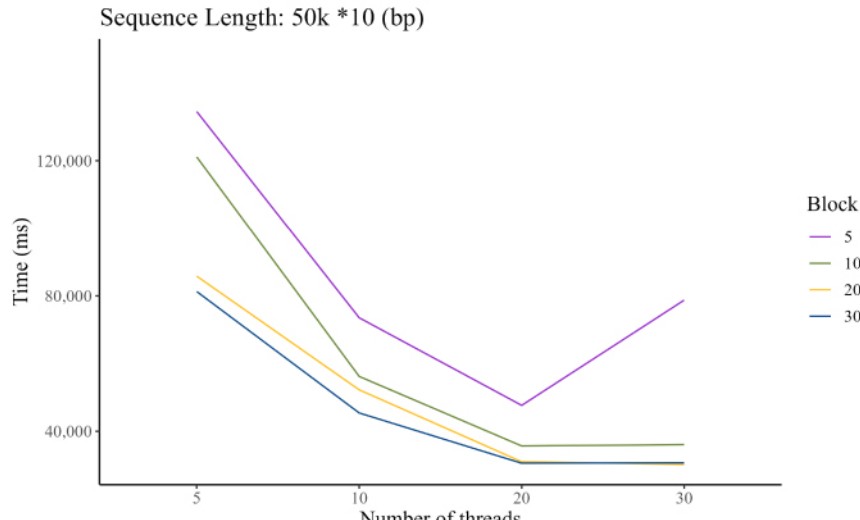

Sequence Length: 50k *10 (bp)

**Figure 6.** Time spent on the alignment of two 200k bases sequences (A) and ten 50k bases sequences (B) in the TSTA algorithm for different block sizes and different numbers of threads.

sequence alignment (200k bases) and multiple sequence alignment (10× 150k bases). The peak speed was achieved when *B* was set to 30 and *T* was set to 30.

It is worth noting that Figure 6 distinctly illustrates a sudden increase in time when *B* equals 5 and *T* equals 30. At this juncture, there is a higher number of threads and smaller blocks, resulting in frequent thread switching. Additionally, the intricate computational preparation of each block counteracts the speedup performance. Furthermore, it is noteworthy that the reduction in time spent is not directly proportional to the increase in block size or the number of threads. While employing a larger block with more threads may

**Table 1.** Time (ms) comparison under the pairwise sequence alignment algorithms.[a]

| Length of sequence | | PSA-time (ms) | | | | | | | |
|---|---|---|---|---|---|---|---|---|---|
| | | Real | | | Simulated | | | | |
| | | 1k | 10k | 50k | 10k | 20k | 50k | 100k | 200k |
| No-traceback | TSTA | 44 | 74 | 204 | 28 | 55 | 176 | 279 | 978 |
| | TSTA-CPU | 15 | 187 | 2,228 | 179 | 658 | 2,235 | 6,353 | 24,476 |
| | seqan_wave | 65 | 199 | 630 | 191 | 290 | 588 | 1,184 | 3,696 |
| | seqan_wave-CPU | 528 | 3,001 | 15,137 | 3,211 | 6,051 | 14,594 | 32,139 | 85,056 |
| | parasail | 149 | 228 | 1,081 | 284 | 409 | 1,452 | error | error |
| | parasail-CPU | 101 | 1,492 | 6,627 | 1,760 | 4,667 | 6,669 | error | error |
| Traceback | TSTA | 49 | 238 | 2,191 | 248 | 480 | 1,846 | 6,001 | 24,385 |
| | TSTA-CPU | 23 | 700 | 15,282 | 716 | 2,622 | 15,117 | 59,848 | 254,249 |
| | bsalign | 11 | 147 | 1,962 | 86 | 316 | 1,925 | 7,659 | 51,147 |
| | parasail | 156 | 580 | 10,576 | 742 | 1,968 | 12,031 | error | error |
| | parasail-CPU | 351 | 6,154 | 16,054 | 5,884 | 7,378 | 16,316 | error | error |

[a] TSTA and parasail were tested separately in two modes with and without backtracking, respectively. "error" indicates that the method could not complete the alignment on the dataset due to memory overflow.

marginally expedite execution time, the improvement in speed is often only a matter of seconds or milliseconds. Consequently, utilizing a larger block or more threads results in an even lower acceleration ratio. In Figure 6, the execution time for a $B/T$ ratio of 30/30 is only marginally shorter than the combined times for 30/20 and 20/30. Consequently, the values of parameters $B$ and $T$ were not escalated continuously. In our alignment approach, both $B$ and $T$ were set to 30.

## Evaluating time under real data

The PSA and MSA were assessed on real and simulated datasets, respectively, as delineated in Tables 1 and 2. In handling real datasets, TSTA demonstrated several advantages across all three datasets. In the "no backtracking" mode, the TSTA algorithm emerged as the fastest, showcasing performance that was 1.47, 2.68, and 3.08 times superior to seqan_wave, and 3.38, 3.08, and 5.29 times greater than parasail, respectively. When operating in "backtracking" mode, the TSTA algorithm proved to be the second most efficient, closely approaching the performance of the bsalign algorithm with a 50k bases dataset. Furthermore, TSTA demonstrated reduced CPU time and decreased thread utilization in the 1k bases dataset, while manifesting elevated values in the other two datasets. In multiple sequence alignment, TSTA showcased the highest speed in the 50k bases dataset, surpassing the SPOA method by 4.22 and 5.02 times and bsalign by 1.68 and 2.61 times in the two datasets, respectively. However, in the 10k bases dataset, its speed was slower than with bsalign, with a trade-off observed with abPOA, which outperformed SPOA. Moreover, the TSTA algorithm exhibited the lowest processing speed among the 1k bases dataset. Overall, in the 50k bases dataset, the TSTA method manifested the highest speed.

## Evaluating time at different lengths in a simulated dataset

We assessed the performance of all algorithms using five categories of sequence lengths for PSA and four categories of sequence lengths for MSA. In PSA, under the "no backtracking" mode, TSTA demonstrated superior real-time and CPU performance compared to other algorithms across all five datasets. In the "backtracking" mode, TSTA ranked second only to bsalign when the sequence length was equal to or less than 20k bases. However, TSTA emerged as the fastest when the sequence length equaled or exceeded 50k bases, while parasail failed to perform the matching task when the sequence length reached or exceeded

**Table 2.** Time (ms) comparison under multiple sequence alignment algorithms.[a]

| | MSA-time (ms) | | | | | | | | | | | | | |
|---|---|---|---|---|---|---|---|---|---|---|---|---|---|---|
| | Real | | | | | | Simulated | | | | | | | |
| Length of sequence | 1k | | 10k | | 50k | | 5k | | 10k | | 20k | | 50k | |
| Number of sequences | 5× | 10× | 5× | 10× | 5× | 10× | 5× | 10× | 5× | 10× | 5× | 10× | 5× | 10× |
| TSTA | 88 | 185 | 1,165 | 3,943 | 14,044 | 36,438 | 443 | 967 | 902 | 2,192 | 2,256 | 5,550 | 8,751 | 19,424 |
| TSTA-CPU | 127 | 380 | 7,772 | 34,578 | 205,142 | 571,643 | 1,802 | 5,029 | 6,535 | 19,635 | 24,476 | 69,349 | 130,293 | 320,236 |
| SPOA | 57 | 78 | 2,177 | 6,436 | 59,404 | 183,219 | 311 | 701 | 1,855 | 4,722 | 7,606 | 19,468 | 50,003 | 128,677 |
| abPOA | 49 | 66 | 1,599 | 3,214 | error | error | 169 | 269 | 820 | 1,726 | 4,478 | 6,305 | error | error |
| bsalign | 61 | 129 | 1,025 | 3,883 | 23,643 | 95,287 | 249 | 736 | 816 | 2,640 | 2,848 | 10,390 | 17,744 | 61,879 |

[a]Time (ms) comparison under the MSA algorithm. Only TSTA uses threads, so only TSTA records CPU time.

**Table 3.** Time (min) comparison under the 60× 100k bases alignment algorithms.

| | MSA-time (min) | | | | | | | | | |
|---|---|---|---|---|---|---|---|---|---|---|
| Real | 41 | 44 | 43 | 42 | 40 | 24 | 25 | 24 | 25 | 25 |
| CPU | 572 | 577 | 583 | 581 | 568 | 602 | 615 | 608 | 611 | 617 |

100k bases. The advantages of TSTA became more apparent with longer sequences. At a sequence length of 100k bases, TSTA was 1.27 times faster than bsalign, and at a length of 200k bases, it accelerated by 2.09 times compared to bsalign. Regarding MSA, TSTA performed at an intermediate level when the sequence length was less than or equal to 10k bases. Nevertheless, TSTA emerged as the fastest when the sequence length equaled or exceeded 20k bases, consistent with the trend observed in PSA. While abPOA encountered limitations in processing sequence lengths of 50k bases, it failed to exhibit a favorable speed trend across sequences ranging from short to lengthy.

To test our algorithm's ability to handle longer sequences in multiple sequence alignments, we performed an MSA test with 60 sequences at 100k bases each (Table 3). Notably, there was a large disparity between the first five alignments and the last five in Table 3. As they were taken from different parts of two groups from the previous pairwise alignments in the 100k bases category, those selected from the same group had similar time expenditures, whereas those from different groups differed greatly due to variations in the construction of the POA graphs.

## More detailed evaluation time

With 30 threads, a more detailed breakdown of the time spent is shown in Tables 4 and 5. Firstly, all times except I/O increased in direct proportion to the number or length of sequences. Additionally, in terms of composition, communication and I/O times made up a very small fraction of the total CPU time, with most of the CPU time being devoted to computation. Notably, the communication time we calculated is the sum of the time spent by all threads when passing through thread locks.

## Contribution of the SIMD and thread

We analyzed whether threading accelerates the program. Table 6 clearly illustrates the time spent under different methods, with the subject being multiple sequence alignments at 5× 20k bases each. We found that using the SIMD technology alone was 2.64 times faster than not using any acceleration techniques. As the number of threads increased from 5 to 30, the speedup rised from 6.69 times to 9.1 times. Hence, threading provided a significant acceleration, but the effect of additional threads diminished considerably.

**Table 4.** Detailed time (ms) comparison under the pairwise sequence alignment algorithms.

| PSA-time (ms) | | | | | |
|---|---|---|---|---|---|
| Length of sequence | 10k | 20k | 50k | 100k | 200k |
| Real | 248 | 480 | 1,846 | 6,001 | 24,385 |
| CPU | 716 | 2,622 | 15,117 | 59,848 | 254,249 |
| Communication | 0.25 | 1.022 | 5.9 | 22.766 | 74.92 |
| I/O | 38.701 | 43.002 | 102.974 | 56.109 | 74.079 |

**Table 5.** Detailed time (ms) comparison under the multiple sequence alignment algorithms.

| MSA-time (ms) | | | | | | | | |
|---|---|---|---|---|---|---|---|---|
| Length of sequence | 5k | | 10k | | 20k | | 50k | |
| Number of sequences | 5× | 10× | 5× | 10× | 5× | 10× | 5× | 10× |
| Real | 443 | 967 | 902 | 2,192 | 2,256 | 5,550 | 8,751 | 19,424 |
| CPU | 1,802 | 5,029 | 6,535 | 19,635 | 24,476 | 69,349 | 130,293 | 320,236 |
| Communication | 0.378 | 0.969 | 1.544 | 5.14 | 8.954 | 23.044 | 38.88 | 100.501 |
| I/O | 39.784 | 28.597 | 10.432 | 1.38 | 1.923 | 3.006 | 40.95 | 15.935 |

**Table 6.** Time consumption under different acceleration conditions.

| 5× 20k bases | | | | |
|---|---|---|---|---|
| | Mode | | Time (ms) | Multiples |
| | No acceleration | | 20,529.6 | 1.0 |
| | Only SIMD | | 7,776 | 2.64 |
| TSTA | Threads | 5 | 3,068 | 6.69 |
| | | 10 | 2,491 | 8.24 |
| | | 20 | 2,332 | 8.8 |
| | | 30 | 2,256 | 9.1 |

**Table 7.** Memory (Mb) comparison under the pairwise sequence alignment algorithms.

| PSA-memory (Mb) | | | | | | | |
|---|---|---|---|---|---|---|---|
| Length of sequence | | 1k | 10k | 20k | 50k | 100k | 200k |
| No-traceback | TSTA | <1 | <1 | <1 | <1 | 1 | 2 |
| | seqan_wave | 5 | 6 | 7 | 8 | 10 | 22 |
| | parasail | 15 | 30 | 27 | 30 | error | error |
| Traceback | TSTA | 4 | 275 | 1,099 | 6,806 | 26,827 | 108,571 |
| | bsalign | 3 | 186 | 732 | 4,540 | 18,181 | 72,291 |
| | parasail | 19 | 392 | 1,470 | 9,113 | error | error |

## Comparison of memory

We conducted a comparative analysis of execution memory costs across various software applications, as presented in Tables 7 and 8. Table 7 illustrates that the backtracking phase incurs the highest level of memory consumption. Although we did not explicitly optimize memory usage, we could conserve it as much as possible. However, our technique led to higher memory consumption compared to bsalign. For example, as shown in Table 8, although TSTA consumes more memory than bsalign, it outperforms SPOA and abPOA in all categories.

## LIMITATIONS

To further validate the effectiveness of TSTA, we conducted tests using several larger datasets. For a dataset containing sequences of 10,000 by 20k bases, TSTA required 890 min, whereas bsalign and spoa took 15 and 2,392 min, respectively. With a dataset comprising 200 sequences of 100k bases each, TSTA completed the processing in 48 min, bsalign took

**Table 8.** Memory (Mb) comparison under the multiple sequence alignment algorithms.

| | MSA-memory (Mb) | | | | | | | | | |
|---|---|---|---|---|---|---|---|---|---|---|
| Length of sequence | 1k | | 5k | | 10k | | 20k | | 50k | |
| Number of sequences | 5× | 10× | 5× | 10× | 5× | 10× | 5× | 10× | 5× | 10× |
| TSTA | 12 | 18 | 127 | 161 | 475 | 605 | 1,847 | 2,324 | 11,283 | 14,255 |
| SPOA | 22 | 30 | 235 | 294 | 1,814 | 2,254 | 7,139 | 8,650 | 44,378 | 53,764 |
| abPOA | 20 | 33 | 378 | 475 | 2,656 | 3,245 | 11,925 | 14,432 | error | error |
| bsalign | 7 | 11 | 87 | 124 | 313 | 402 | 1,288 | 2,131 | 8,584 | 11,326 |

98 min, and spoa was unable to complete the alignment. In a smaller dataset of 50 sequences, each consisting of 200 bases, TSTA processed the data within 12 min, bsalign required 20 min, and spoa once again failed to complete the task. Additionally, we attempted to test sequences with lengths of 100k and 200k bases, but due to memory constraints in our testing environment, all three algorithms were unable to proceed.

From these results, it is evident that TSTA performs well in scenarios involving longer sequences (>50k bases) and a moderate or smaller number of sequences. However, when dealing with a larger number of shorter sequences, TSTA's efficiency falls short of expectations, although it still surpasses the performance of spoa.

## DISCUSSION AND CONCLUSIONS

Employing threads to expedite sequence alignment holds promise for longer sequences but may not be advantageous for shorter ones. In the latter cases, both multithreading and striped SIMD data structures become redundant. Optimization strategies for alignment must strike a balance between speedup and overhead. Furthermore, employing threads for certain intricately designed algorithms may not yield substantial speedups. For instance, in the case of the bsalign algorithm, which also employs the difference and striping methods, reconfiguring it according to the design of our algorithm may disrupt memory accesses and result in unchanged or even inferior outcomes. However, it is noteworthy that a more refined multithreaded design solution could potentially alter this scenario.

Additionally, during the backtracking phase, a lot of memory is used, and accessing the data created during this phase takes a significant amount of time. Thus, optimizing the backtracking stage is critical for future research.

Our findings demonstrate TSTA's efficacy in pairwise sequence alignment and multiple sequence alignment, particularly with long reads, showcasing considerable speed enhancements compared to existing tools. We anticipate that TSTA will offer enhanced value and play a pivotal role in bioinformatics.

## AVAILABILITY OF SOURCE CODE AND REQUIREMENTS

- Project name: TSTA
- Project home page: https://github.com/bxskdh/TSTA
- Operating system(s): Platform independent
- Programming language: C
- License: MIT
- Biotools: biotools:tsta
- RRID:SCR_025750.

## DATA AVAILABILITY

All supporting datasets are available in Figshare [28].

## ABBREVIATIONS

AVX, advanced vector extensions; DP, dynamic programming; DRR, difference recurrence relations; MSA, multiple sequence alignment; POA, partial order alignment; PSA, pairwise sequence alignment; SIMD, Single Instruction Multiple Data; SSE, Streaming SIMD Extensions.

## DECLARATIONS

### Ethics approval and consent to participate

Not applicable.

### Competing interests

The authors declare that they have no competing interests.

### Authors' contributions

PZ, conceptualization; PZ, project administration; PZ and WD, software; PZ, data collection, processing, and application; WD, project coordination; PZ, manuscript writing and figure generation; JL, manuscript review.

### Funding

This work was supported by the National Key Research and Development Program of China (2022YFC3400300 to JR) and the Innovation Program of the Chinese Academy of Agricultural Sciences.

### Acknowledgements

We would like to thank Mr. Deng for his help in testing the algorithm and reviewing the manuscript.

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
