## [Editor Report]

Editor’s AssessmentThe article presents strategies for accelerating sequence alignment using multithreading and SIMD (Single Instruction, Multiple Data) techniques, and introduces a new algorithm called TSTA (Thread and SIMD-Based Trapezoidal Pairwise/Multiple Sequence-Alignment). The Technical Release write-up presenting a detailed description of TSTA's performance in pairwise sequence alignment (PSA) and multiple sequence alignment (MSA), and compares it with various existing alignment algorithms. Demonstrating the performance gains achieved by vectorized SIMD technology and the application of threading. Testing and debugging a few errors, and adding some more background detail, demonstrating it can achieve faster comparison speed. Demonstrating TSTA's efficacy in pairwise sequence alignment and multiple sequence alignment, particularly with long reads, and showcasing considerable speed enhancements compared to existing tools.Editor’s AssessmentThe article presents strategies for accelerating sequence alignment using multithreading and SIMD (Single Instruction, Multiple Data) techniques, and introduces a new algorithm called TSTA (Thread and SIMD-Based Trapezoidal Pairwise/Multiple Sequence-Alignment). The Technical Release write-up presenting a detailed description of TSTA's performance in pairwise sequence alignment (PSA) and multiple sequence alignment (MSA), and compares it with various existing alignment algorithms. Demonstrating the performance gains achieved by vectorized SIMD technology and the application of threading. Testing and debugging a few errors, and adding some more background detail, demonstrating it can achieve faster comparison speed. Demonstrating TSTA's efficacy in pairwise sequence alignment and multiple sequence alignment, particularly with long reads, and showcasing considerable speed enhancements compared to existing tools.

---

## [Reviewer Report]

Indicate in the comments box below whether you are happy with the changes made or if the manuscript is unacceptable.Comments on revised manuscriptThe authors themselves acknowledged that their tool performs well only for moderate or small datasets, which restricts its utility in broader applications that require large-scale sequence analysis. For a journal like GigaByte, which focuses on scalable and impactful bioinformatics tools, this limitation may hinder its relevance for publication.

---

## [Reviewer Report]

Indicate in the comments box below whether you are happy with the changes made or if the manuscript is unacceptable.Comments on revised manuscriptAfter thoroughly reviewing the revised manuscript and testing the TSTA tool, I cannot endorse the manuscript for publication in its current form. I encourage the authors to address the following issues thoroughly and consider re-submitting after significant improvements. Efficiency Concerns: In the context of multiple sequence alignment (MSA), I find that TSTA does not demonstrate a significant advantage in terms of efficiency. I conducted a test with approximately 2G of homologous diploid reads (not too large data), and the tool has been running for around 29 hours. Despite this extensive runtime, the process remains incomplete. This is far from the efficiency one would expect from a tool designed for large-scale sequence alignment. Functionality Issues: There are still unresolved issues with the tool's functionality. The -f parameter does not appear to work as intended, and there are also problems with the -o parameter. Such issues need to be addressed to ensure the tool's reliability and usability.

---

## [Reviewer Report]

Reviewer name and names of any other individual's who aided in reviewerBaoxing SongDo you understand and agree to our policy of having open and named reviews, and having your review included with the published manuscript. (If no, please inform the editor that you cannot review this manuscript.)YesIs the language of sufficient quality?YesPlease add additional comments on language quality to clarify if neededIs there a clear statement of need explaining what problems the software is designed to solve and who the target audience is? YesAdditional CommentsIs the source code available, and has an appropriate Open Source Initiative license <a href="https://opensource.org/licenses" target="_blank">(https://opensource.org/licenses)</a> been assigned to the code?YesAdditional CommentsAs Open Source Software are there guidelines on how to contribute, report issues or seek support on the code?YesAdditional CommentsIs the code executable?YesAdditional CommentsIs installation/deployment sufficiently outlined in the paper and documentation, and does it proceed as outlined?YesAdditional CommentsIs the documentation provided clear and user friendly?YesAdditional CommentsIs there enough clear information in the documentation to install, run and test this tool, including information on where to seek help if required?Additional CommentsIs there a clearly-stated list of dependencies, and is the core functionality of the software documented to a satisfactory level?YesAdditional CommentsHave any claims of performance been sufficiently tested and compared to other commonly-used packages? YesAdditional CommentsIs test data available, either included with the submission or openly available via cited third party sources (e.g. accession numbers, data DOIs)?Additional CommentsAre there (ideally real world) examples demonstrating use of the software? YesAdditional CommentsIs automated testing used or are there manual steps described so that the functionality of the software can be verified?Additional CommentsAny Additional Overall Comments to the AuthorZong et al. implemented a TSTA package that integrated the difference method, the stripe method, SIMD, and multiple threading approaches to perform efficient sequence alignments. The TSTA toolkit could conduct pairwise and multiple sequence alignments. The memory cost of TSTA is comparable with the most efficient one. Overall, TSTA is a good package, and the manuscript is well-written. While I have a few suggestions: 1) The minimap2 should be mentioned in the section on "difference recurrence relation." It has a much broader range of users and implemented an algorithm that is slightly different from the one by Suzuki, etc. 2) The striped SIMD is also implemented in reads mappers, such as BWA. 3) Page 14, line 215 "1k bps", line 227 "1000 kbps", line 230 and table1 "100k". They should be consistent. 4) In Table 4, I am not sure I understood the second and third lines correctly. Please clarify. 5) I tried to compile TSTA from the source code. To compile the package, I had to copy 'seqio.h' into the 'msa' and 'psa' folders. Please fix it.RecommendationMinor Revisions

---

## [Reviewer Report]

Reviewer name and names of any other individual's who aided in reviewerYuansheng LiuDo you understand and agree to our policy of having open and named reviews, and having your review included with the published manuscript. (If no, please inform the editor that you cannot review this manuscript.)YesIs the language of sufficient quality?YesPlease add additional comments on language quality to clarify if neededIs there a clear statement of need explaining what problems the software is designed to solve and who the target audience is? YesAdditional CommentsIs the source code available, and has an appropriate Open Source Initiative license <a href="https://opensource.org/licenses" target="_blank">(https://opensource.org/licenses)</a> been assigned to the code?YesAdditional CommentsAs Open Source Software are there guidelines on how to contribute, report issues or seek support on the code?YesAdditional CommentsIs the code executable?YesAdditional CommentsI do not test.Is installation/deployment sufficiently outlined in the paper and documentation, and does it proceed as outlined?YesAdditional CommentsIs the documentation provided clear and user friendly?YesAdditional CommentsIs there enough clear information in the documentation to install, run and test this tool, including information on where to seek help if required?YesAdditional CommentsIs there a clearly-stated list of dependencies, and is the core functionality of the software documented to a satisfactory level?YesAdditional CommentsHave any claims of performance been sufficiently tested and compared to other commonly-used packages? YesAdditional CommentsIs test data available, either included with the submission or openly available via cited third party sources (e.g. accession numbers, data DOIs)?Additional CommentsAre there (ideally real world) examples demonstrating use of the software? NoAdditional CommentsIs automated testing used or are there manual steps described so that the functionality of the software can be verified?Additional CommentsAny Additional Overall Comments to the AuthorThe article explores strategies for accelerating sequence alignment using multithreading and SIMD (Single Instruction, Multiple Data) techniques, and introduces a new algorithm called TSTA. The paper provides a detailed description of TSTA's performance in pairwise sequence alignment (PSA) and multiple sequence alignment (MSA), and compares it with various existing alignment algorithms. Experimental results indicate that TSTA demonstrates significant speed advantages, particularly when handling long sequences and in the no-backtracking mode. However, the experiments on MSA are limited by the experimental environment, which does not fully address the needs of current sequencing technologies concerning long reads and depth. Specifically, the low number of sequences in MSA does not meet the requirements for downstream genomic analysis applications. While the algorithm is highly innovative, its performance on short sequences and during the backtracking phase still requires optimization. 1. In line 7, the TSTA algorithm utilizes vector-level and thread-level parallelism to accelerate pairwise and multiple sequence alignment. Why are there no experiments designed specifically to evaluate the global alignment performance of TSTA with vector-level parallelism? Or are there any other experimental designs that demonstrate the improved performance of TSTA when vector-level parallelism is employed? 2. In line 149, is the Active-F method used by the TSTA algorithm contributing to the excessive memory usage and access time overhead observed during the iterative process of PSA? Are there better optimization strategies from this perspective? If not, why does TSTA incur higher time costs in traceback as shown in Table 1? Why does bsalign result in lower time consumption? 3. Can you provide the time breakdown for each part of the parallel computation in TSTA for PSA (including at least CPU computation overhead, communication overhead, and I/O overhead) to clarify if there will be significant communication overhead issues with larger datasets and more threads? 4. Table 2 shows that both real and simulated datasets have issues with insufficient depth and short reads. In real MSA processes, it is common to encounter comparisons with depth over 60X and lengths exceeding 100 kbps for long reads. The results under the current experimental conditions seem to perform poorly for such data scenarios. Can you address this? 5. Gene data often includes repetitive regions that affect the accuracy of alignment algorithms. Can you design experiments to verify how TSTA performs in aligning long repetitive regions? Specifically, how accurately does TSTA align sequences in such regions compared to other methods? 6. Besides repetitive regions, sequencing errors produced by ONT R10 chips can also impact alignment accuracy. Alignment algorithms used in genome correction often struggle to detect such errors. How does TSTA handle such issues during MSA? Can the algorithm be designed to address these sequencing errors more effectively?RecommendationMajor Revisions